# Selection and transmission of the gut microbiome alone can shift mammalian behavior

Taichi A. Suzuki[1,2], Akbuğa-Schön Tanja[1], Jillian L. Waters[1], Dennis Jakob[1], Dai Long Vu [3], Mallory A. Ballinger[4], Sara C. Di Rienzi [5,6], Hao Chang[7], Ivan E. de Araujo[7,8], Alexander V. Tyakht[1] & Ruth E. Ley [1,9] ✉

Animals live in partnership with their gut microbiota, and these microbial communities often shift when hosts adapt to new environments. While it is well known that the microbiome can influence traits ranging from metabolism to behavior, a key question remains unresolved: can host traits under natural selection be transmitted solely through the microbiome, without changes to the host genome? Here we experimentally demonstrate that selection on a behavioral trait in mice significantly shifts the host trait over time through microbiome transmission alone. We first identify locomotor activity as transmissible through the gut microbiome, using fecal transfers from wild-derived mouse strains into germ-free male recipients. Building on this, we carry out four rounds of one-sided microbiome selection, serially transferring microbiomes from low-activity donors to independently bred male germ-free mice. Only this selection line, not the randomly chosen control line, shows a decrease in locomotion toward the end of the experiment. Reduced activity is linked to enrichment of *Lactobacillus* and its metabolite indolelactic acid, and administration of either alone is sufficient to suppress locomotion. These findings demonstrate that microbiome selection and transmission can shape mammalian behavior, independent of host genomic evolution. Our work highlights the role of microbiome-mediated trait inheritance in shaping host ecology and evolution.

Natural selection acts on phenotypic variation, which is influenced by genetic and environmental factors including the microbiome. Host-associated microbiomes are increasingly recognized as important sources of host phenotypic variation[1–8]. Microbiomes influence host developmental processes and niche space, with potential impacts on their host species' evolutionary trajectory[1–7]. Within a single host genotype, variation in the microbiome can also produce different phenotypes. In examples of microbe-driven host plasticity, aphids with the same genotypes can exhibit different parasitoid defense[9], body colors[10], and thermotolerance[11], due to variability in their symbiotic bacteria alone. In such relatively simple systems, it is clear that the symbionts can shape adaptive host phenotypes[12]. However, it remains

[1]Department of Microbiome Sciences, Max Planck Institute for Biology, Tübingen, Germany. [2]College of Health Solutions and Biodesign Institute for Health Through Microbiomes, Arizona State University, Tempe, AZ, USA. [3]Mass Spectrometry Facility, Max Planck Institute for Biology, Tübingen, Germany. [4]Department of Ecology and Evolutionary Biology, Cornell University, Ithaca, NY, USA. [5]Center for Advanced Biotechnology and Medicine, Rutgers University, Piscataway, NJ, USA. [6]Department of Neuroscience and Cell Biology, Rutgers University, Piscataway, NJ, USA. [7]Department of Neuroscience, Icahn School of Medicine at Mount Sinai, New York, NY, USA. [8]Department of Body-Brain Cybernetics, Max Planck Institute for Biological Cybernetics, Tübingen, Germany. [9]Cluster of Excellence EXC 2124 Controlling Microbes to Fight Infections, University of Tübingen, Tübingen, Germany. ✉e-mail: rley@tuebingen.mpg.de

unclear whether microbiomes in more complex vertebrate systems can similarly contribute to host phenotypic variation as the basis for natural selection[13,14].

Experimental evolution studies in vertebrates suggest that the microbiome may play a role in mediating adaptation alongside changes in the host genome. In many of these studies, microbiome composition significantly differs between selection lines and unselected control lines when hosts are selected for traits such as locomotion[15–18], predatory behavior[16], and herbivorous diet[16]. Given that microbiome transplant experiments using germ-free mice can often transfer donor phenotypes to recipients[19–22], it is commonly assumed that microbiome changes during selection experiments partially contribute to host trait variation. Supporting this idea, in mouse lines selectively bred for high-running performance, antibiotic depletion of the microbiome reduces running ability, while unselected control lines are largely unaffected[23]. However, it remains challenging to disentangle whether host phenotypic responses to selection are driven by changes in the host genome, the microbiome, or both. This highlights the need for experimental approaches that can isolate the role of the microbiome in driving host phenotypic changes, independent of host genomic evolution.

A powerful approach to disentangle how the microbiome responds to host-level selection independent of host evolution is to use one-sided host-microbiome selection experiments[13,24]. In these experiments, microbiomes are selected from donor hosts exhibiting a desired trait and serially transferred to independently generated germ-free recipients. When performed in plants, such experiments have demonstrated that selected soil microbiomes can induce changes in host traits over time, including plant biomass[25], flowering time[26], drought tolerance[27], and salt tolerance[28]. Notably, these changes in plant traits occurred relatively quickly, within five rounds of transfer. While these plant studies demonstrate the feasibility of microbiome-driven trait selection, they raise the question of whether similar selection dynamics occur in animals.

Selection on microbiome-influenced host traits in animals requires that the microbiome component driving the trait is present over multiple generations of host[1–7]. In species of insects that vertically transmit their endosymbionts with high fidelity, this condition is met[5]. In the more complex setting of mammalian gut microbiomes, transmission routes are mixed, but there is evidence of vertical transmission in an increasing number of microbial species[29–31]. Moreover, certain microbial species in human gut microbiomes show evidence of transmission within genetically related individuals over thousands of generations[32]. Such long-term codiversification patterns are observed widely across primates[33,34], and other vertebrates[35]. These observations across species provide evidence that vertical inheritance of specific members of the gut microbial community may be widespread in mammals. This sets the stage for microbially conferred host phenotypes to be passed intergenerationally and potentially selected upon, in which case phenotype-conferring microbes would act similarly to host genes. Despite this potential, few studies have directly tested whether host phenotypes under selection in animals can be shaped by multi-generational microbial inheritance alone, independent of host genomic changes.

To our knowledge, the only such study in animals was conducted using *Drosophila melanogaster*[36]. Microbiomes from the fastest enclosing flies were transferred into new media with sterile eggs from stock flies. However, despite a well-controlled design with replicates, the developmental time of the selection lines did not significantly differ from those of the random control lines after four rounds of transfer[36]. Thus, there remains a critical gap in demonstrating that one-sided host-microbiome selection can alter microbiome-mediated host traits in animals.

Here we performed a one-sided host-microbe selection experiment on gut microbiome-driven traits in the house mouse (*Mus musculus domesticus*). We show that selection and transmission of the gut microbiome are sufficient to alter a host behavioral trait. By transferring gut communities from low-activity donor mice into germ-free recipients over four rounds of transfer, we demonstrate that the microbiome selection line exhibits a stronger decrease in locomotor activity than the control line. Reduced activity is associated with enrichment of *Lactobacillus* and its metabolite indolelactic acid, and administering either alone is enough to reduce locomotion. These results demonstrate that microbiome-mediated host traits can be selected and transmitted, driving changes in host behavior over time, independently of host genetic changes.

## Results

### Determining donors and target traits in one-sided host-microbiome selection experiment

We first assessed wild-derived *Mus musculus* inbred lines as potential donors for a one-sided host-microbe selection experiment (Fig. 1A) because (i) intraspecific microbiomes from wild-derived inbred lines mitigate losses compared to interspecific fecal transfer[37] and (ii) a diverse wild-derived starting community is expected to improve the selection efficiency over domesticated microbiomes that have lost ancestral variation[13,38–40]. We used the well-characterized and highly inbred line C57BL/6NTac as recipient germ-free mice to control for the effects of host genetic variability on trait variation[13,38].

We assessed two inbred mouse lines as microbiome donors: (i) the SAR line, originating in Saratoga Springs, New York, USA, and (ii) the MAN line, originating in Manaus, Brazil[41] (Fig. 1A). We characterized variation in 16 physical and behavioral traits in both lines (Fig. 1B), including locomotion-related traits measured with an automated cage system[42]. We observed significant differences between male SAR and MAN lines for body weight, tail length, and locomotion behavior (Fig. 1B–E; Figure S1), as previously reported[41,43,44]. Traits with the largest mean difference for SAR versus MAN lines were 'total activity' (i.e., sum of all movements), and 'distance traveled' (i.e., sum of walking and running; Fig. 1B, E).

Next, we assessed the same set of traits in male C57BL/6NTac mice after conventionalization via fecal transfer from SAR and MAN donors (Fig. 1A; Methods). Overall, we observed that the transfer of microbiomes from SAR and MAN donors transmitted the differences observed in donor lines to the recipients, albeit to a greater extent for the behavioral traits compared to the morphological traits (Fig. 1B–E, Supplementary Data 1, Fig. S1). While different host genotype–microbiome combinations can yield distinct interactions[14,45,46], the consistent phenotype transfer highlights the robustness of the microbial effect. Tail length was the only trait that deviated from this pattern, showing a significant difference in the opposite direction of the donor phenotype (Fig. 1D), potentially due to the greater energy extraction capacity of the SAR microbiome compared to that of MAN[47] (See Supplementary Information). Among the assessed traits, distance traveled showed the strongest phenocopying via microbiome transfer (Fig. 1B, E). Distance traveled significantly differed between SAR and MAN donors (Wilcoxon test uncorrected $p = 0.0000063$) and between SAR and MAN recipients (Wilcoxon test uncorrected $p = 0.031$) with the largest effect sizes within donors (269%) and within recipients (15%) (Fig. 1B, E). The differences in distance traveled remained significant after accounting for body weight at inoculation and batch effects (See Supplementary Information, Figs. S2, S3, Likelihood ratio test $p < 0.05$). Our findings are consistent with quantitative genetic studies that generally show lower genetic contribution and higher environmental influence on behavioral traits compared to morphological traits[48].

Finally, we confirmed that the microbiomes of the SAR and MAN mice differed, and these differences in donor-mouse microbiomes could be partially transferred to recipient mice (PERMANOVA, $F = 15.5$, $p < 0.001$) (Fig. 1F, Supplementary Data 2, 3). Based on these combined results, we chose distance traveled as the microbiome-induced trait to select for in the one-sided host-microbiome selection experiment.

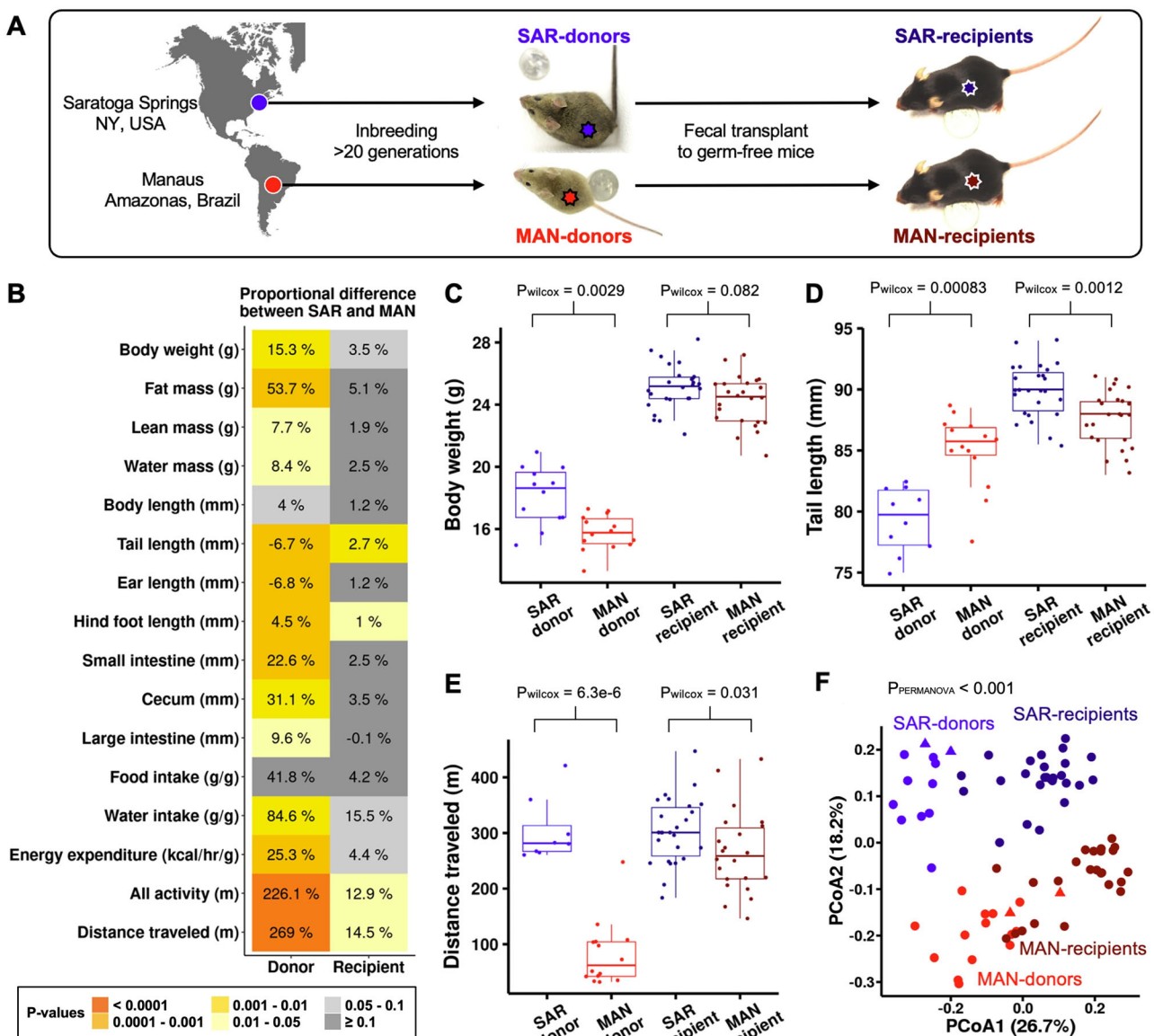

**Fig. 1 | Host traits affected by microbiome. A** Experimental design. Feces from two wild-derived inbred lines of house mice (SAR donors and MAN donors) were transferred to the cages of germ-free C57BL/6NTac (SAR recipients and MAN recipients). **B** Proportional differences (%) of traits between SAR donors (*n* = 10) and MAN donors (*n* = 14) and between SAR recipients (*n* = 26) and MAN recipients (*n* = 24), all single-housed males. Positive values show greater trait values in SAR compared to MAN and negative values show the opposite. Food intake, water intake, and energy expenditure are corrected for body mass. Colors indicate Wilcoxon rank sum test uncorrected *p*-values. Examples of donor and recipient traits: **C** Body weight, (**D**) Tail length, and (**E**) Distance traveled. Sample sizes in **C–E** follow those in panel B. Box plots: box = median + IQR; whiskers = 1.5 × IQR. Pwilcox: Wilcoxon rank sum test uncorrected *p*-values (all tests two-sided). **F** Mouse fecal microbiome between-sample diversity displayed as Principal Coordinates Analysis using Bray-Curtis distances. PERMANOVA results and the percentage of variation explained by PCoA1 and PCoA2 are indicated. Circles represent fecal microbiomes from each mouse; triangles are the fecal samples used for the transplant experiment at the start and end of the experiment.

## Microbiome transmission drives mouse activity behavior

We set up a one-sided host-microbiome selection experiment (Fig. 2A; Methods) using the SAR line as microbiome donors and with the goal of reducing distance traveled. We chose to reduce (rather than increase) distance traveled as this reflects the direction of adaptation that has occurred in nature; ancestral mice from higher latitudes were introduced to lower latitudes, where activity levels are observed to be lower[49,50]. We set up two lines: (i) the selection line, in which the two male mice with the least distance traveled after 24 h at 5–6 weeks of age were chosen as donors for the next round, and (ii) the control line, in which the two mouse donors were selected at random (Fig. 2A). Recipients were inoculated at 3–4 weeks of age through coprophagy, with each round lasting 2 weeks, for 4 rounds (N0-N4; Fig. 2A).

We observed that selection for low mouse activity behavior via transfer of the microbiome significantly reduced the median distance traveled over the whole course of the experiment (Fig. 2). As expected, in the N0 transfer time point, the control and selection lines had similar median distances traveled (*p* > 0.05; Fig. 2B; Supplementary Data 4). Correcting for starting weight and batch effects, which were mainly caused by litter availability (Figs. S2, S3, see Supplementary Information), the median distance traveled decreased over three and four rounds of transfer in the selection lines (N0-N3: Wilcoxon test *P* = 0.017; N0-N4: Wilcoxon test *P* = 0.058), but not in the control lines or when using the raw data uncorrected for these covariates (Fig. 2B). We obtained a similar result from model comparisons based on likelihood ratio tests (Fig. 2C, Supplementary Data 4). These results

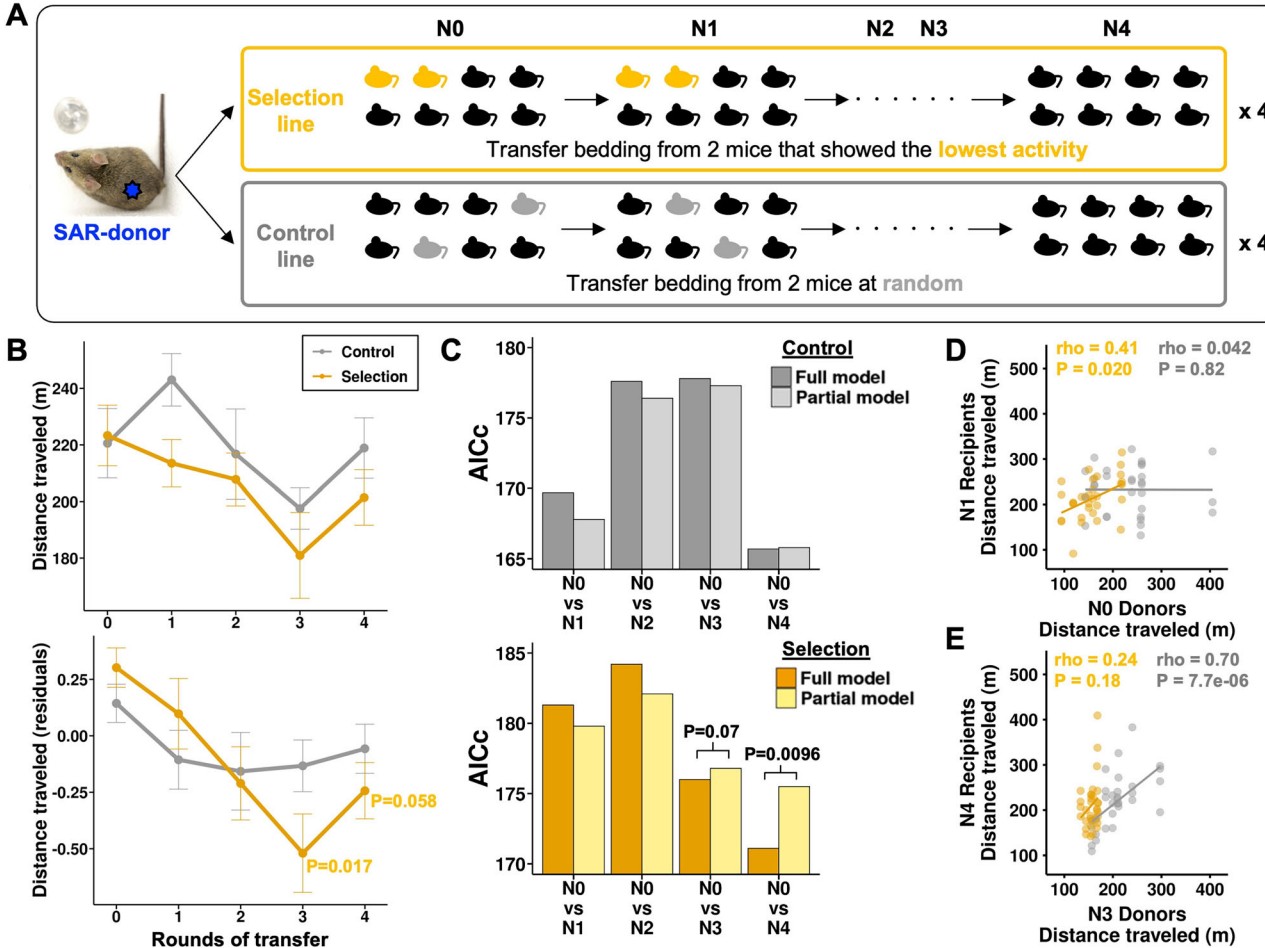

**Fig. 2 | Selection and transmission of microbiome drive activity behavior.**
**A** Experimental design. SAR-donor mouse feces were placed in germ-free recipient mouse cages. Recipients were split into two 8-mouse groups. Selection line (yellow): selection for the least distance traveled; Control line (gray): selected randomly. Process repeated four rounds ($n = 311$ germ-free mice). **B** Medians and standard errors of distance traveled for four selection lines ($n = 155$) and four control lines ($n = 155$). Upper panel: raw data in meters; Lower panel: residuals accounting for initial body weight and batch effects. $P$-values of Wilcoxon rank sum test comparing before (N0) and after selection (N1, N2, N3, and N4) are indicated:

$p > 0.1$ are not displayed (all tests two-sided). **C** Results of model comparisons of control lines (upper panel) and selection lines (lower panel). The full model includes the distance traveled as the response variable; rounds of transfer and body weight at inoculation as fixed and batch as random effects. The Partial model excludes rounds of transfer. AICc is plotted and $p$-values are based on the likelihood ratio test. Spearman's rho correlations of distance traveled between N0 donors and N1 recipients (**D**) and between N3 donors and N4 recipients (**E**). Data points represent individual donor-recipient pairs used in fecal microbiome transfer. Control line (gray); Selection line (yellow).

indicate that selection and microbiome transmission were effective at reducing the distance traveled.

Next, we tested whether the distance traveled for one round of transfer was perpetuated to the next through microbiome transmission. We observed a significant positive correlation of distance traveled between the donors and recipients using all data (one round to the next; rho = 0.193, $p = 0.0021$, Fig. S4). Interestingly, the temporal patterns of phenocopying differed between the selection and control lines (Fig. 2D, E, Fig. S4). The selection line showed a significant phenocopying of distance traveled at the start of the experiment (N1-N2 rounds of transfer, rho = 0.41, $p = 0.020$, Fig. 2D), but not for the rest of the time points (Fig. S4). In contrast, the control line showed significant phenocopying of distance traveled only at the end of the experiment (N3-N4 rounds of transfer, rho = 0.70, $p = 7.7e{-}06$, Figs. 2E, S4). Overall, our results support a significant phenocopying of distance traveled through microbiome transfer, consistent with the findings from the wild-derived mice in this study (Fig. 1E) and a previous study using Diversity Outbred donors[51]. Moreover, our observations are consistent with theoretical predictions[24,52], that selection should deplete the microbiome variation affecting distance traveled at the start of the experiment in the selection lines, while stochastic

processes are more likely to maintain this variation through the end of the experiment in the control lines.

## Signatures of selection on the microbiome composition

Our experiments reveal that both treatment and rounds of transfer altered the gut microbiome. Analysis of the cecal microbiome diversity showed that microbiome richness (alpha diversity) was similar for the two treatments and decreased with rounds of transfer (Fig. 3A, S5; Supplementary Data 5). The gut microbiomes of mice from the control and selection lines started off with similar composition (beta diversity; N0) then diverged significantly from the N0 and with each other over the rounds of transfer (Fig. 3B, C, Supplementary Data 6). We detected a significant effect of round of transfer on overall microbiome composition (PERMANOVA, $F = 10.2$, $p < 0.001$, Fig. 3B), which was stronger than the effect of treatment (selection vs. control), although treatment still showed a significant but weaker association with Bray-Curtis dissimilarity (PERMANOVA, $F = 2.8$, $p = 0.006$, Fig. 3B). Microbiome differences between control and selection lines were only evident at the later rounds of transfer (N0, N1, and N2, PERMANOVA, $p > 0.05$; N3 and N4, PERMANOVA, $p < 0.05$, Fig. 3C). Overall, the parallel shifts observed in both host traits and microbiome diversity between control

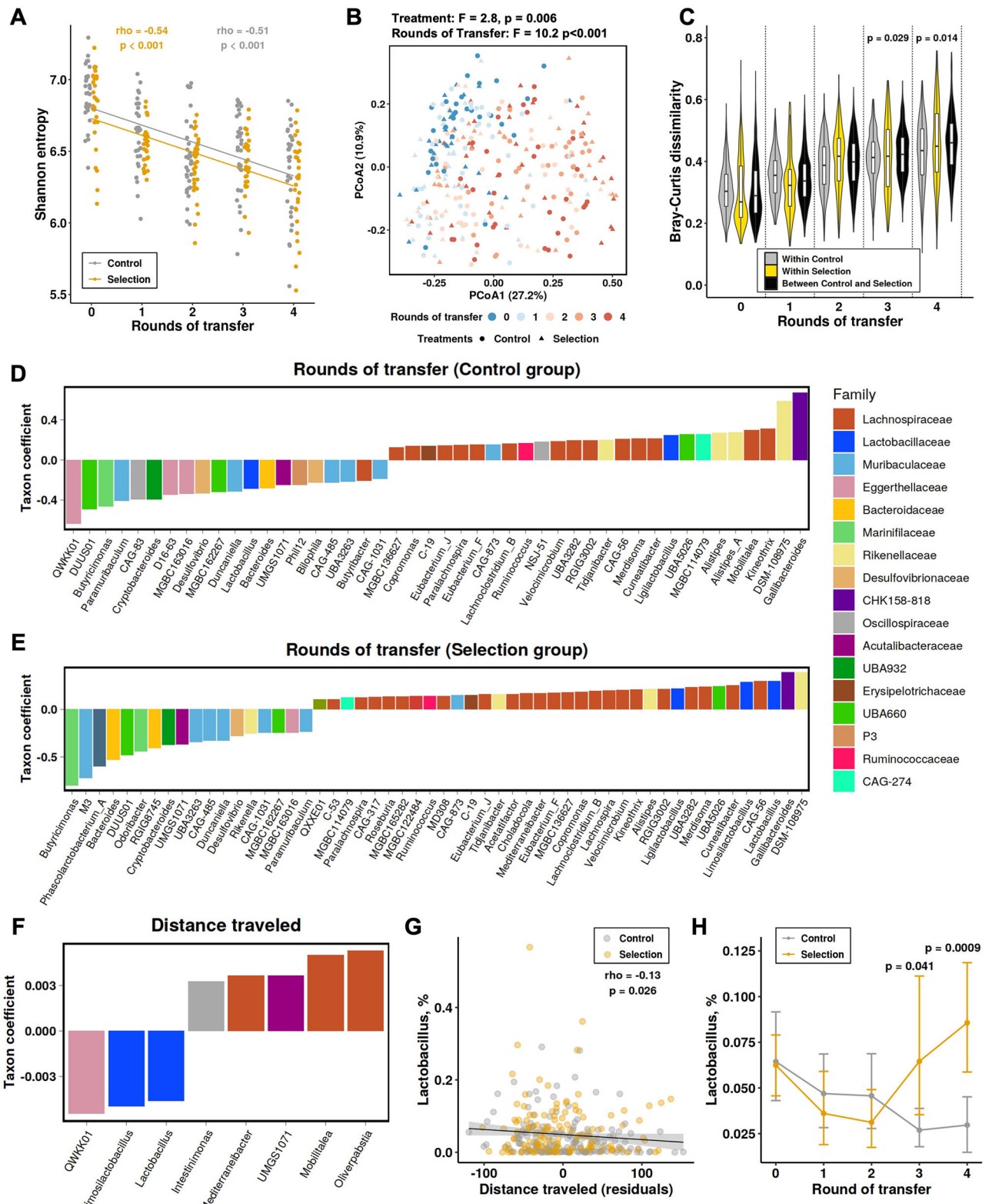

and selection lines at later rounds of transfer (Figs. 2B, C and 3C) support the notion that changes in behavior were influenced by changes in the microbiome. The observed loss of richness and the divergence between treatments over time are consistent with the stochastic loss of microbes caused by community bottlenecking and divergent ecosystem-level selection[13,24,53].

To identify specific microbial taxa whose ratios of relative abundances associated with rounds of transfer or treatment, we conducted

a compositionally aware nearest balance (NB) analysis (Methods). The NB method robustly identifies associations by evaluating ratios of bacterial taxa rather than their individual abundance values (percentages) and aggregating them into a single fraction (i.e., balance)[54]. We observed an increase in the balance numerator for the following taxa with rounds of transfer (i.e., these taxa increased; adjusted for treatment effects): *Gallibacteroides* (family CHK158-818 of the Bacteroidales), *DSM-108975* (Rikenellaceae), and many genera belonging to

**Fig. 3 | Microbiome changes in the selection experiment. A** Shannon entropy changes across rounds of transfer, Spearman rho and *p*-value are shown. **B** PCoA plot of Bray-Curtis dissimilarity colored by rounds of transfer, with different shapes corresponding to treatments; PERMANOVA results are shown. **C** Bray-Curtis dissimilarity within Control lines ($n = 152$, gray), within Selection lines ($n = 151$, yellow), and between Control and Selection lines ($n = 303$, black); *p*-values are based on pairwise PERMANOVA testing the effect of treatment on Bray-Curtis dissimilarity within groups (control-control and selection-selection) versus between groups (control-selection) within each transfer round. Violin box plots: violin = density of all values; box = median + IQR; whiskers = 1.5 × IQR. **D** Nearest balance (NB) associated with rounds of transfer in the Control group ($n = 152$), showing taxon coefficients and per-sample computed values, with microbial genera having positive values being enriched and those with negative values depleted across rounds. **E** Similar NB analysis for the Selection group ($n = 151$). **F** NB associated with distance traveled. **G** Correlation between distance traveled and *Lactobacillus* relative abundance with 95% CI (shaded). **H** Changes in *Lactobacillus* over time for the Control ($n = 152$) and Selection ($n = 151$) lines, presented as means with bootstrapped 95% CI; P-values indicate Wilcoxon tests comparing Selection and Control lines within rounds of transfer (all tests two-sided). Colors represent Selection (yellow) and Control lines (gray) in (**A**, **C**, **G**, and **H**).

Lachnospiraceae and Rikenellaceae (Fig. 3D, S6A; Supplementary Data 7). On the other hand, the balance denominator taxa (decreased with rounds of transfer) included *QWKK01* (hereafter, *Enterorhabdus* as per GTDB Release 220), *Butyricimonas*, Muribaculaceae and other families. This indicates that the serial transfers alone influenced microbiome composition over four rounds of transfer.

Selection for low activity resulted in an increase in *Enterorhabdus*, *Lactobacillus* and *Limosilactobacillus*, and four other genera, along with a reduction in proportions of *M3* (Muribaculaceae family) and eight diverse genera, when controlling for the effects of rounds of transfer (Fig. S6B). The influence of selection on the microbiome is well illustrated by distinct changes in microbial genera only in the selection line (Fig. 3E), but not in the control line (Fig. 3D). The described patterns were generally confirmed by the NB analysis at both the species level and the metagenome assembled genome (MAG) level (see Supplementary Data 8–24). The effects of selection on microbial community composition became apparent as the balance values shifted with rounds of transfer, starting with initial similarity between control and selection lines (N0) and leading to divergence in later selection rounds (Fig. S6C). These findings suggest that the effects of selection and rounds of transfer on the microbiome were largely distinct. Yet, for overlapping taxa they strongly counteracted each other, with selection suppressing *Gallibacteroides* and *DSM-108975*, along with various Lachnospiraceae members, while enriching for *Enterorhabdus*, *Lactobacillus* and *Limosilactobacillus*, *Paramuribaculum*, and *D16-63* (Eggerthellaceae family) (Fig. S6A, B).

### Microbial taxa associated with mouse locomotion

We computed the NB associated with distance traveled, controlling for the body weight at inoculation (see Supplementary Information, Fig. S2, 3) and accounting for the same variables above. Distance traveled was negatively associated with the genera *Enterorhabdus*, *Lactobacillus*, and *Limosilactobacillus*, and positively associated with the enrichment of *Intestinimonas* and several genera within the *Lachnospiraceae* (Fig. 3F, S7A). These associations mirror the enrichment of *Lactobacillus* and *Limosilactobacillus* in the selection lines (Fig. S6B). Although a significant correlation between the NB value and distance traveled was observed in both control and selection lines, it was slightly stronger in control lines (Fig. S7B), consistent with selection diminishing variability in distance traveled within selection lines relative to control lines (Fig. 2D, E).

Further focusing on *Lactobacillus* as a key contributing taxon, we observed significant negative correlation of its relative abundance with the adjusted distance traveled across all data (Fig. 3G). When analyzing the correlation in each selection round separately, all but one round exhibited negative slopes, with significance reached in 2 of the 5 rounds ($p < 0.05$). The most pronounced and almost equal correlations occurred in the two final selection rounds (N3: rho = 0.28, $p = 0.029$; N4: rho = −0.27, $p = 0.035$) (Fig. S8). Additionally, *Lactobacillus* stood out among the top three taxa most negatively associated with distance traveled (Fig. 3F) due to higher relative abundance in the selection lines compared to the control lines in later rounds of transfer (Fig. 3H, S9), aligning with observed changes in host behavior (Fig. 2B, C).

### Indolelactic acid (ILA) associates with selection, locomotion, and microbiome

We measured serum concentrations at the start (N0) and end (N4) of the experiment for 12 targeted metabolites previously linked to microbiome modulation and behavioral effects in animals[42]. Following the correction of metabolite levels and distance traveled for the body weight at inoculation, rounds of transfer significantly explained the overall variation in metabolites ($R^2 = 0.067$, $p < 0.0001$), whereas selection (interaction of rounds of transfer and treatment) and distance traveled did not significantly contribute (Fig. 4A). Corrected concentrations of two metabolites decreased (cortisol, corticosterone), and one increased (thyroxine), over the course of the experiment (N0-N4, $p < 0.05$, Fig. 4B).

ILA and thyroxine showed a significant negative correlation with distance traveled (Spearman rho = −0.18 and −0.19, uncorrected $p = 0.0451$ and 0.035, respectively; Fig. 4C and Fig. S10A; Supplementary Data 25), while for GABA it was positive (rho = 0.19, $p = 0.0393$; Fig. S10B). Given the *Lactobacillus'* capacity to catabolize tryptophan to ILA[55], we focused on the ILA, a microbial metabolite of tryptophan implicated in gut-brain axis communication[56–58]. Search for microbial genera whose ratios were associated with levels of ILA using NB yielded a balance of *Lactobacillus* and *Limosilactobacillus* (along with two other genera) to *Bilophila* and *Desulfovibrio* genera (with four others; Fig. 4D, S11). A similar analysis at the species level revealed three closely related species in the genus *Lactobacillus* had the strongest positive association with ILA (Fig. S12A, B). We confirmed the contribution of the leading species (*Lactobacillus johnsonii)* with a MAG-based balance analysis (Fig. S12C, D). The result is further supported by the significant positive correlation between the relative abundance of the genus *Lactobacillus* and ILA (rho = 0.35, $p = 1.2E-04$, Fig. 4E). These findings are consistent with the previously reported associations between *Lactobacillus* abundance, tryptophan metabolites, and hyperactivity in rodents and humans[56–60].

### Administration of *Lactobacillus johnsonii* and ILA to the gut reduces locomotion

Given that our NB analysis identified *L. johnsonii* as the leading and the most abundant species of the *Lactobacillus* genus associated with reduced activity (Fig. S12A, C), and that this species is known to produce ILA[55], we next directly tested the effect of *L. johnsonii* and ILA on the activity behavior of mice. First, we verified that *L. johnsonii* strain LJ0, that we had previously isolated from C57BL/6J mice[61] produced lactate. In pure *L. johnsonii* cultures, ILA was detected at $2.07 \pm 0.13\,\mu g/mL$ in conditioned medium and $25.3 \pm 2.5\,ng/mg$ in dry cell mass (mean ± SD, $n = 5$).

We administered this strain to conventionally-raised C57BL/6J mice via oral gavage. Administration of *L. johnsonii* LJ0 resulted in significantly reduced distance traveled in an open field test relative to the vehicle control treatment (Fig. 5A, B; two sample *t*-test, $p = 0.011$; 8 and 12 mice in the saline and *L. johnsonii* groups, respectively), as was speed of the mice (Fig. 5C; two2 sample t-test, $p = 0.014$) with no changes to food intake or body weight (Fig. 5D, E; two2 sample t-test, $p > 0.05$).

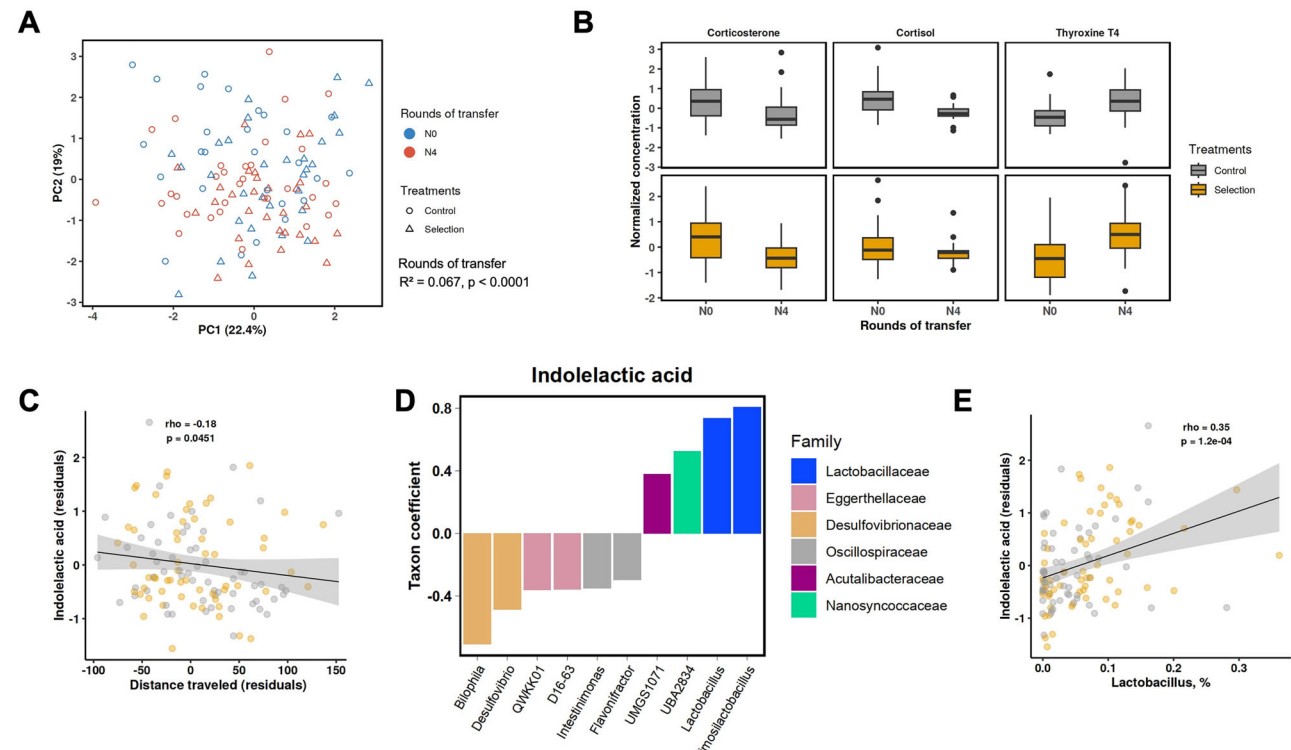

**Fig. 4 | Targeted metabolite changes in the selection experiment.** The levels had been corrected for the body weight at inoculation. **A** Metabolome ordination plot based on 12 metabolites, with linear mixed model results for rounds of transfer displayed. **B** Three metabolites significantly different between the start (N0) and end (N4) of the experiment based on linear mixed-effects model (all tests two-sided). Box plots: box = median + IQR; whiskers = 1.5 × IQR. **C** Correlation between ILA and distance traveled with 95% CI (shaded). **D** Nearest balance associated with the ILA. **E** Correlation between ILA and *Lactobacillus* with 95% CI (shaded). Colors indicate Selection ($n = 62$, yellow) and Control ($n = 59$, gray) lines (**B**, **C**, and **E**).

To assess the effect of ILA directly in the small intestine, we administered it to the duodenum of a separate group of mice via an implanted catheter (Methods). Again, we observed a significant reduction in activity for the ILA-administered compared to the control mice (Fig. 5F, G; $p = 0.0009$; 8 and 5 mice in the saline and ILA groups, respectively) and reduction in speed (Fig. 5H; $p = 0.0009$) with no changes to food intake or weight (Fig. 5I, J; $p > 0.05$). These results support a direct effect of ILA produced by *Lactobacillus* spp. as causative in lowering the activity of mice.

## Discussion

The results of our study suggest that components of the microbiome can serve as a non-nuclear mechanism for generating host phenotypic variation that can be acted on by selection. Mammals have evolved parental care, facilitating microbial transmission from parents to offspring. Vertical transmission across generations has been observed to be widespread in different mammals[29–31,62] and can lead to host-microbial codiversification[32,34]. Like host genes, vertically transmitted microbes that influence host phenotype have the potential to contribute to the heritable variation upon which natural selection can act[7]. The microbiome-mediated adaptive plasticity demonstrated here is distinct from other transgenerational models of plasticity that require genetic variation in the host genome or epigenetic modifications such as DNA methylation[63]. The novelty in our study lies in experimentally demonstrating that selection on a host trait can lead to changes in that same trait over time purely through microbiome transmission, without any genetic evolution in the host. The results provide experimental support for theoretical models[24] and highlight the potential for microbiome engineering[13] in mouse models, advancing our understanding of ecosystem-level selection in complex mammalian microbiomes[6,7,64,65].

Pioneering studies on one-sided host-microbiome selection experiments have faced criticism for their lack of control treatments, replicates, and documentation of microbiome changes[13]. We specifically addressed these limitations in our experimental design through: (i) identifying key host traits influenced by the microbiome prior to the selection experiment, (ii) including a random control line[13,36], (iii) conducting four biological replicates per treatment, and (iv) characterizing changes to the microbiome and host metabolites. Despite these improvements, the effect size of host trait changes in response to selection (change in median distance traveled in meters: N0 to N3 = −18.9%, N0 to N4 = −9.8%) was smaller compared to previous studies in plants[13,25–27]. This may be partly due to our conservative design, in which the control line was established by random selection and happened to include low-activity individuals, thereby reducing the contrast in selection pressure between the selection and control lines. Including two selection lines in opposite directions (e.g., high and low activity) alongside a random control line, increasing the number of individuals per transfer round, and extending the number of rounds of transfer would enhance the statistical power to detect the effects of selection.

Aspects of the experimental design may limit the applicability of our findings to natural populations. Coprophagy was used to simulate natural microbiome transmission, but we acknowledge that high-fidelity transmission of a complex microbiome from a single individual is unrealistic in nature[5,30]. In addition, the absence of microbiomes during the germ-free recipients' early development (0–3 weeks old) likely affected immune development and microbiome assembly[66]. To address these issues, future studies should investigate the resilience of the selected microbiome by exposing animals to external microbial sources or by utilizing recipients with existing microbiomes instead of germ-free mice. Although the host genome was kept constant in our

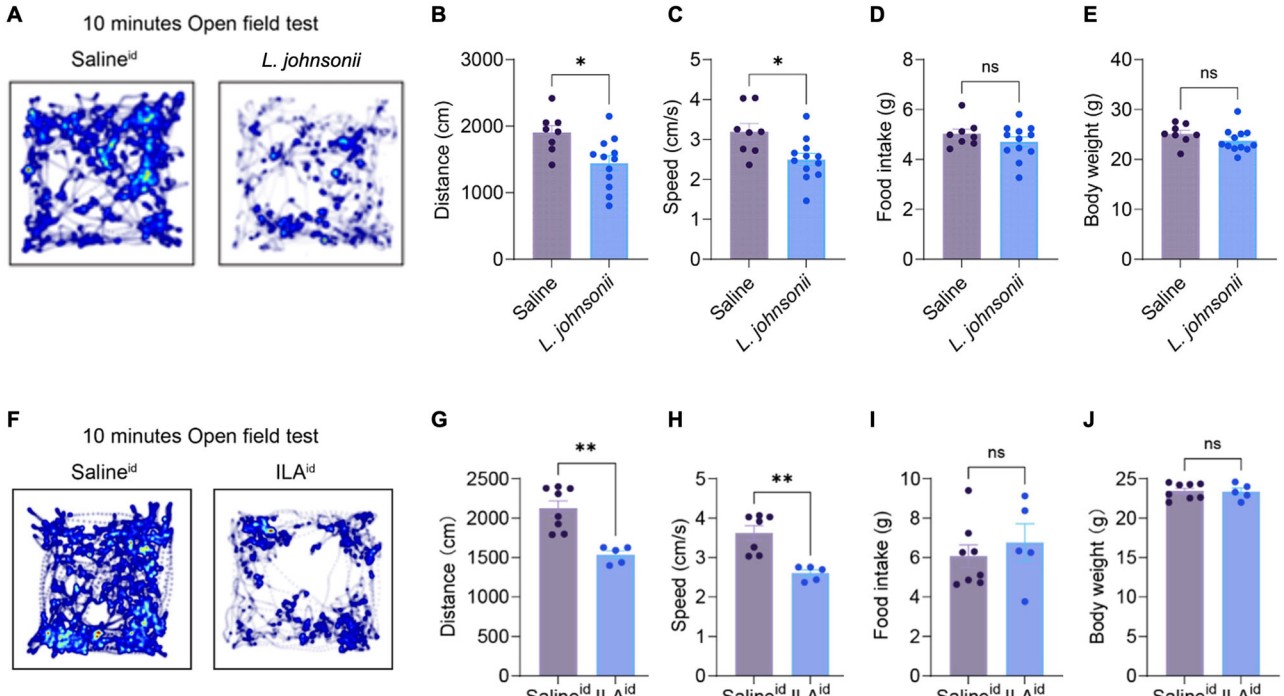

**Fig. 5 | Administration of *Lactobacillus johnsonii* or ILA reduces activity in mice.** **A**–**E** Open-field tests for mice with administration of saline ($n = 8$) or *L. johnsonii* solution ($n = 12$, 200 μL of ~$10^9$ cells/mL for 4 days) by oral gavage. **A** Heatmap of 10 min open-field tests. **B** Total distance. **C** Average speed. **D** Daily food intake.

**E** Body weights. **F**–**J**: same as **A**–**E** but for the mice with intraduodenal administration of saline ($n = 8$) or ILA ($n = 5$, 40 mg/kg). Bar plots represent means ± SEM. Two-sample t-test was used: **$p < 0.001$, *$p < 0.05$, ns $p > 0.05$ (all tests two-sided).

experiment, we anticipate larger and faster changes in selected traits when both the host genome and heritable microbiome components respond to selection[14].

Selection on low-activity mice and transfer of their microbiomes had the effect of reducing levels of *Lactobacillus* spp. in the gut and ILA in circulation. Tryptophan metabolism is increasingly implicated in the communication between the gut microbiota and central nervous system[67]. The ILA is a bacterial metabolite of tryptophan with known immune modulatory effects through the aryl hydrocarbon receptor AhR[68–70]. These anti-inflammatory effects extend to the central nervous system[71,72]. *Lactobacillus* abundance has been linked to animal behavior in both rodents and humans[56–58,73–75] and to memory formation in bees via AhR[76]. Our results suggest that the tryptophan metabolism of *Lactobacillus* spp., and subsequent ILA availability, may act on the brain to modulate activity behaviors[77]. This provides a mechanistic hypothesis for the targets of selection within complex microbial communities[5].

We acknowledge that behavior is highly context dependent. The open field test used in the follow-up experiment, for instance, does not directly replicate the behavioral context of the selection experiment. Variables such as time of day, age, room conditions, and handling can all influence behavioral outcomes. Thus, the open field assay should not be viewed as a direct test of causality. Rather, the selection experiment guided us to identify a specific microbe (*Lactobacillus johnsonii*) and metabolite (ILA) that independently induce similar behavioral changes. In a broader context, although we did not measure distance traveled in germ-free mice under sterile conditions, the values observed in the initial (median 250–300 m per 24 h, Fig. 1E) and selection experiment (median 180–250 m per 24 h, Fig. 2B) are similar to or lower than the published measurements for germ-free C57BL/6 mice using the same automated cage system (Promethion, Sable Systems)[78]. Reduced activity levels of inoculated germ-free mice compared to sterile germ-free mice are consistent with previous reports[79]. Further studies will be necessary to evaluate the ecological relevance of the locomotor behavior targeted in this study.

The gut microbiome encodes a vast array of genes that expands the genomic and phenotypic capacity of the host, influencing how animals respond to natural selection. Microbes that persist across host generations are particularly important, as they are more likely to drive host-microbial coevolution compared to frequently exchanged microbes among individuals or populations[5]. Our results demonstrate that specific microbiome components can be subject to selection for their effects on host phenotype, independently of changes in the host genome. This highlights that even within complex mammalian gut communities, certain microbes can act as non-nuclear mechanisms driving host ecology and evolution. These insights not only have practical relevance for improving traits in livestock and biomedical contexts[13], but also provide a framework for understanding how microbiomes may shape host adaptation under environmental and climate change.

## Methods

### Animals

All procedures adhered to German and U.S. regulations (Regierungspräsidium Tübingen EB 02/20 M, EB 04/19 M; Mount Sinai IACUC-2018-0041). Two wild-derived inbred lines, Manaus, Brazil (MAN), and Saratoga Springs, USA (SAR), originally maintained at the University of California Berkeley, were maintained in individually ventilated cages at the Max Planck Institute for Biology since 2018. Germ-free C57BL/6NTac mice were bred in isolators and screened monthly for contaminants. All germ-free mice used in this study were male. Animals were maintained on standard chow (Altromin 1314) *ad libitum*. Separate experiments used specific-pathogen-free C57BL/6J males (8–12 weeks, Mount Sinai facilities), maintained on standard chow (PicoLab 5053) *ad libitum*.

### Fecal-transplant experiment

Fresh feces (10–15 pellets) from 28-week-old MAN or SAR breeder females were placed into cages of newly weaned germ-free C57BL/6NTac males (3–4 weeks). Recipients were housed two per cage and

phenotyped at 8 weeks for behavior and metabolism in Promethion chambers (Sable Systems), body composition by EchoMRI, and standard morphology. Donor cohorts (14 MAN, 10 SAR) and recipient cohorts (24 MAN-recipients, 26 SAR-recipients) were processed in three biological replicates over seven months. Indirect calorimetry, activity, and distance traveled were extracted at 5-min resolution and analyzed with ExpeData. Normality was assessed, and statistical comparisons used Wilcoxon rank-sum tests (aka Mann–Whitney U-test) and nested mixed-effects models that corrected for body weight at inoculation and batch effects with FDR correction (see Supplementary Information).

## One-sided host-microbiome selection experiment

Feces from a single SAR female were placed in 16 cages of singly housed, newly weaned (3–4 weeks of age) germ-free males (N0). These 16 cages were evenly split into two groups: a selection line and a control line. After two weeks, feces from two of the lowest-activity individuals in the selection line and from two randomly chosen individuals in the control line were used to inoculate the next set of 16 cages, each containing independently bred germ-free males. This serial transfer was repeated four times (N0–N4) in four independent replicates, totaling 311 mice. Behavioral and physiological traits were measured 14 days post-inoculation (5–6 weeks of age). Serum sampling was performed in N0 and N4 animals for targeted metabolomics (see below). Distance-travel data were analyzed with Wilcoxon rank-sum tests and nested mixed-effects models. Microbiome covariates were evaluated with PERMANOVA and nearest-balance analysis.

## *Lactobacillus johnsonii* and ILA inoculation experiments

ILA production of *Lactobacillus johnsonii* strain LJ10 (GenBank assembly GCA_002156645.1) was confirmed by LC-MS/MS quantification of ILA in both culture supernatant and cell extracts. Eight- to ten-week-old C57BL/6 J males received LJ10 ($\approx 10^9$ CFU in 200 µl saline) or vehicle by oral gavage for four consecutive days, followed by a 10-min open-field test analyzed with EthoVision XT. For ILA administration, 8-week-old mice bearing duodenal catheters were infused with ILA (40 mg kg$^{-1}$ day$^{-1}$) or saline for four days before the same behavioral assay. Two-sample t-test was used.

## Targeted metabolomics

Serum from N0 and N4 mice was analyzed for pre-selected 12 gut–brain metabolites[42] using HPLC (UltiMate 3000, Thermo Fisher, USA) coupled to high-resolution MS (Impact II, Bruker, Germany). Four compounds followed a published protocol[42]; eight used a C18 Kinetex column (150 × 2.1 mm, 40 °C) with a 0.4 ml min$^{-1}$ gradient containing 0.1% formic acid. Collision energies were 15–35 eV; ILA-d$_5$ served as internal standard. Peaks were integrated using Skyline (v.21.1). Metabolite levels were adjusted for body weight and modeled by linear mixed-effects regression with FDR correction. Details of sample preparation, controls, settings, and annotation are provided in the published protocol[42] and Supplemental Information.

## Metagenomic sequencing and analysis

Cecal DNA (PowerSoil kit) was tagmented with Nextera Tn5 (1 ng input) and dual-indexed by 14-cycle PCR. Size-selected (400–700 bp) libraries were pooled and sequenced on an Illumina HiSeq 3000 (2 × 150 bp). Reads were adapter-trimmed (Skewer), quality-filtered (bbduk), and host-depleted (bbmap). Taxonomic profiles were generated with Kraken2/Bracken; functional profiles with HUMAnN3 using a Struo2-built GTDB r207 database. Samples were rarefied to 150,000 reads for diversity analyses in QIIME2. Non-rarefied datasets were also profiled against a custom MAG database (KrakenUniq). Community variation was tested by PERMANOVA on Aitchison distances (9999 permutations); taxa–trait associations used nearest-balance analysis with centered log-ratio-transformed counts (see details in Supplementary Information).

## Statistics & reproducibility

For microbiome compositional analyses, one sample (R3N5T290) was excluded from all downstream analyses due to a low number of reads. For nearest-balance analyses, seven outlier samples (outside median ± 3 SD) based on body weight at inoculation or distance traveled were excluded. For metabolomic analyses, three outlier samples were excluded. Cage positions and littermates of germ-free mice were randomized. The initial fecal transfer experiment from two wild-derived donors was performed across three independent replicates, and the one-sided selection experiment was conducted for four rounds across four independent replicates. Each biological replicate (e.g., control and selection lines) was always run in parallel. No statistical method was used to predetermine sample size. Investigators were not blinded to allocation during experiments or outcome assessment. See details in Supplementary Information.

## Reporting summary

Further information on research design is available in the Nature Portfolio Reporting Summary linked to this article.

## Data availability

The gut metagenomes and metagenome-assembled genomes data generated in this study have been deposited in the European Nucleotide Archive database under accession code PRJEB83173. The metabolomic data generated in this study have been deposited in the MassIVE repository under data identifiers: MSV000097923, MSV000097925, MSV000097928 [https://massive.ucsd.edu/ProteoSAFe/dataset.jsp?task=085984b5b0a241769a32ee9facf5dec9, https://massive.ucsd.edu/ProteoSAFe/dataset.jsp?task=3955f76b9597404583c5a0a49a800b69, https://massive.ucsd.edu/ProteoSAFe/dataset.jsp?task=62d02218f2f1488fa33c8f080c7f0531]. The mouse phenotypic data generated in this study are provided in the Supplementary Data file.

## Code availability

The code used in data analysis is provided at GitHub (https://github.com/leylabmpi/sel_trans_behavior/) and Zendo[80].

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

## Acknowledgements

We thank Annkatrin Geysel, Matthias Neuscheler, Julia Leibßle, Silke Dauser, Heike Budde, and Stacey Heaver for their assistance. This work was funded by the Max Planck Society and the European Research Council (ERC) under the European Union's Horizon 2020 research and innovation program Grant agreement (SilentFlame ID 101142834).

## Author contributions

Conceptualization: R.E.L., T.A.S. Data curation: T.A.S., A.V.T., D.J., D.L.V., H.C. Formal analysis: T.A.S., A.V.T., H.C. Funding acquisition: R.E.L., IdA Investigation: T.A.S., A.V.T., H.C. Methodology: T.A.S., T.A.-S., J.W., D.J., D.L.V., S.C.D., M.A.B., S.C.R. Project administration: R.E.L., IdA Software: A.V.T. Supervision: R.E.L., IdA Visualization: T.A.S., A.V.T., H.C. Writing – original draft: T.A.S., R.E.L., A.V.T.

## Funding

## Competing interests

The authors declare no competing interests.
