## [Transparent Peer Review file · Nature Communications]

Selection and transmission of the gut microbiome alone can shift mammalian behavior

Corresponding Author: Dr Ruth Ley

Version 0:

Reviewer comments:

Reviewer #1

(Remarks to the Author)

In this manuscript it is addressed whether host phenotypic responses to selection in mice can be driven by the microbiome alone. Host traits transmissible via fecal microbiome transplantations from two phenotypically distinct wild-derived inbred mouse strains into a third, germ-free inbred line were characterized, and locomotor activity occurred as the apparent. Using locomotor activity as a trait to select (selection line), the gut microbiome was transferred from low-activity mice to independently bred germ-free mice in four transfer rounds. This selection line was compared to a control line, in which microbiome selection was random. Microbiome transfer in the selection line, but not in the control line, led to a significant reduction in locomotor activity in recipients. The reduced activity was associated with increased *Lactobacillus* abundance and their metabolite indolelactate. Administration of *Lactobacillus johnsonii* or indolelactate alone reduced locomotion. It is concluded that selection and transmission of microbes can modulate mammalian phenotypic change independently of selection on the host genome.

The organization of the manuscript is a bit weird, as it is written with the first part without a subheading, and then some subheadings. All the methods are put into the supplementary. I think there should be a distinct introduction, and there should be a shorter methods section, while it is OK to have a supplementary methods file with more details.

I think the referral to plant microbiomics in both the first part and the discussion is a bit awkward. We do not need to explain to readers of Nature Communications that plants do not have pregnancies, birth and foster care.

Apart from this the manuscript is well written and easy to follow.

• What are the noteworthy results?

The results are noteworthy, i.e. activity is to a wide extent microbiomically and not so much genetically driven. This is not fully novel, as there are several papers, some cited and some not cited in this manuscript, showing a strong microbiomic impact on behaviour, although the locomotor angle to a certain extent is new.

• Will the work be of significance to the field and related fields? How does it compare to the established literature? If the work is not original, please provide relevant references.

The work will be of significance to the field and related fields, as it very thoroughly documents the relation between microbiome and locomotor activity. Especially the transplantations are valuable, both because wild mouse microbiota is applied, and because such studies often do not do transplantations. Wild mouse microbiota is more diverse than the SPF microbiota, and, therefore, this is more relevant for humans, and it is probably also easier to reveal any differences. There could have been more references to microbiota works showing relation behaviour in mice, such as Bercik et al 2011. 'The Intestinal Microbiota Affect Central Levels of Brain-Derived Neurotropic Factor and Behavior in Mice', *Gastroenterology*, 141: 599-609.e3 and Bravo et al 2011. 'Ingestion of *Lactobacillus* strain regulates emotional behavior and central GABA receptor expression in a mouse via the vagus nerve', *Proc.Natl.Acad.Sci.U.S.A.*

• Does the work support the conclusions and claims, or is additional evidence needed?

I think in general the conclusions are well supported.

• Are there any flaws in the data analysis, interpretation and conclusions? Do these prohibit publication or require revision?

I have difficulties understanding why Wilcoxon's Signed-Rank test is applied, as the data sets in it do not seem to be paired. In that case it should just be a Mann-Whitney U-test. Also, it is not clear why a non-parametric test is applied, because it is not described if the data were subjected to any normality and equality tests.

- Is the methodology sound? Does the work meet the expected standards in your field?
It probably does, but we need more information on the animals (see below).

- Is there enough detail provided in the methods for the work to be reproduced?
It is not clear whether the intra-duodenal catheterization was done under anesthesia and if so, which one. It is also not clear whether analgesics were given afterwards. If not done under anesthesia and eventually some analgesics the procedure does not seem to be in accordance with the EU directive.

The information on housing of the animals is very sparse, and we know nothing about enrichment, cage types, bedding etc.

What does it mean that health status was normal and how was this assessed? Normally, laboratory mice are free of a range of listed pathogens, but these mice were either of wild mouse origin and not subjected to rederivation, or they were germ-free, so even though I agree that the wild mouse derived mice are more natural, they do not seem very normal for laboratory animal facilities.

Germ-free is written as both germfree and germ-free. Correct MESH term is germ-free.

ILA is not properly defined in the text.

Reviewer #2

(Remarks to the Author)

In this original research, the authors test the hypothesis that phenotypic evolution of a host can be mediated by through microbiome transfer. The researchers first established a set of traits that were transferable via the microbiome and then chose one of these traits for a one-sided selection study to determine if host behavior could be modified through the microbiome. The research is an elegant, integrative study that is impressively replicated. The methods are sound, and in many cases exceed typical standards for work in this area. The results reveal at least one possible target (single microbe) of the selection experiments and delves deeply into mechanism through metabolite analysis and subsequent experiments. The research demonstrates that selection on the microbiome can significantly alter host phenotype independent of host genetics.

Major questions/comments

My central questions/comments revolve around the phenotype of distance traveled.

In the initial microbiome transfer experiments to the germ-free mice (results in Figure 1), how was it ascertained that it was the microbiome composition that altered the phenotype of the germ-free mice, and not the genetic background of the germ-free mice? Fig 1C suggests that germ-free mice, with either microbiome (MAN or SAR) travel more than the MAN line. How active are germ-free mice when germ-free? Are the results reported proportional to the baseline state of germ-free mice? Here, I am trying to understand how the microbial transfers changed the activity of the germ-free state?

How much less active were the mice within the experimental treatments, and what was the differential between the selection lines and the control lines? That is, what was the effect size on overall movement? It looks like 10% by the end of the experiment. It would be useful for the reader to know the effect.

What is the fidelity of distance traveled? How much variation does an individual display among measurements?

Minor comments:

Pg 145 should this say "selection lines?"

What is plotted in Fig 2B, D? is it the mean of the 4 selection lines? Or individual performance? Please specify.

Line 146 "or when using the raw data" what does this mean?

Fig 3 F -what is the insert? It is too blurry to see what is in the insert.

227- specify direction seems like this is "reduction" in distance traveled

230- what does "this" refer to? The pattern? The outcome? Same for 233

242: "top 3 negatively contributing taxa". This is confusing, please revise

344: Does *Lactobacilli* need to be italicized or written as lower case? What is the convention?

363: "microbes that persist across generations"- host generations?

Reviewer #3

(Remarks to the Author)

The manuscript by Suzuki et al. provides valuable experimental evidence that microbiome selection alone can influence host traits in mammals, representing an important proof-of-principle for microbiome-mediated inheritance mechanisms. However, the central behavioral phenotype appears to rely heavily on statistical adjustments for confounding variables like body weight, with small effect sizes that required a high n and are visible after complex modeling rather than in raw data. While the microbiome and metabolite findings are compelling and the mechanistic follow-up with *Lactobacillus johnsonii* strengthens the work, the modest effect sizes and dependence on statistical correction of the behavioral data give me pause.

With major revisions addressing the statistical robustness of the behavioral claims and clearer presentation of effect sizes alongside significance testing, this could represent an important contribution to our understanding of microbiome-mediated evolution.

- When were microbiome samples analyzed?

- Why did you choose to use the open field test as a behavior marker of phenotype transmission?

- It appears that a large difference in behavior is required in the donors (effect size 269%) to then pass to recipients (effect size of 15%). Does this indicate that only extreme differences may be transferable by microbiota transplant?

- Not quite sure I understand why you chose the SAR line as microbiome donors to reduce distance traveled, as they were the group with the higher distance traveled? Could you not have used the MAN donors to SAR recipients?

- Why did you choose 2 mice at random for the control donors? If the random mice were either high or low in locomotor activity this could be a potential confounder.

- How was coprophagy microbiota transfers confirmed? It is difficult to assess dose and frequency using this method. It appears that some microbes were indeed transferred but it is difficult to say if this led to recipients having closely similar microbiome compositions to the donors.

- Single housing in itself is a stressor that can affect mouse behavior.

- It is not clear to me whether the behavior data in figure 2B is more driven by microbiota transfer or other variables. Did you expect the total distance traveled for control and selection to move in a similar direction? It looks like body weight was more different than the locomotor behavior, and it is well known that heavier mice typically move less. This makes me think that perhaps the microbes influence behavior through weight change rather than modulating other pathways that impact behavior.

- Can you identify what the batch effects are?

- It is difficult for me to see the difference in Bray-Curtis dissimilarity across control and selection groups. Can you explain what is being compared by the pairwise PERMANOVA p-value that is reported? I am concerned that the differences are being driven by a large n rather than biological relevance and that treatment explains much less variance than rounds of transfer.

- Is locomotion speed necessary to include in figure 5? This is just distance divided by the standard 10 min test.

- The study focuses on Lactobacillus but microbiome changes were complex - other taxa may contribute significantly. That being said, I appreciate that authors followed up initial findings and tested both one of the differentially abundant microbes and metabolites.

- This is essentially a single experiment (though with internal replication). Independent replication would strengthen conclusions.

- Why were the metabolomic analyses focused to 12 targeted compounds?

- Were all data subjected to FDR testing?

- Were there any outliers removed?

- In Figure 1B both dark yellow and light grey are referred to as $P < 0.01$. Please define what the gray boxes represent in the figure or figure legend.

- The use of "non-nuclear" in the one sentence summary is somewhat confusing terminology. I think I understand what the authors are asserting but perhaps something along the lines of "extra-genomic mechanism" or "microbiome-mediated inheritance" would be easier for readers to understand.

Version 1:

Reviewer comments:

Reviewer #1

(Remarks to the Author)

In general, the authors have followed my advice, and we have also had more information on the animals. I have no further comments.

Reviewer #2

(Remarks to the Author)

Thank you for your responses to my comments. I realized my first comment was not as clear as it could have been. It is reproduced below::

In the initial microbiome transfer experiments to the germ-free mice (results in Figure 1), how was it ascertained that it was the microbiome composition that altered the phenotype of the germ-free mice, and not the genetic background of the germ-free mice?

What I was attempting to ask was how is it known that it was the different microbiomes that caused the differences in phenotype and not just the genetics of the host (or a potential interaction with host genes). Since the comparison was between hosts with different microbiomes, and not a comparison to germ-free mice, how are the authors certain that the phenotype is caused by the microbiome per se and not the underlying genetics of the host. That does the phenotype differ from the germ free state?

Reviewer #3

(Remarks to the Author)

Despite my initial concerns about modest effect sizes and statistical dependence, the authors adequately justified their conservative experimental design and demonstrated that effects are visible in raw data, not just after corrections. The independent validation through *Lactobacillus* experiments and the rigorous use of random controls (rather than extreme phenotypes) strengthen the biological interpretation. While independent replication would be ideal, this represents the first experimental demonstration of microbiome-mediated trait inheritance in mammals, a significant conceptual advance regardless of effect magnitude. This work opens exciting new avenues for understanding how microbial communities can drive evolutionary change and may have important implications for microbiome engineering approaches. I recommend to accept this manuscript for publication.

Response to Reviewers:

Author's response to reviewers in blue.

Reviewer #1 (Remarks to the Author):

In this manuscript it is addressed whether host phenotypic responses to selection in mice can be driven by the microbiome alone. Host traits transmissible via fecal microbiome transplantations from two phenotypically distinct wild-derived inbred mouse strains into a third, germ-free inbred line were characterized, and locomotor activity occurred as the apparent. Using locomotor activity as a trait to select (selection line), the gut microbiome was transferred from low-activity mice to independently bred germ-free mice in four transfer rounds. This selection line was compared to a control line, in which microbiome selection was random. Microbiome transfer in the selection line, but not in the control line, led to a significant reduction in locomotor activity in recipients. The reduced activity was associated with increased *Lactobacillus* abundance and their metabolite indolelactate. Administration of *Lactobacillus johnsonii* or indolelactate alone reduced locomotion. It is concluded that selection and transmission of microbes can modulate mammalian phenotypic change independently of selection on the host genome.

Thank you for your review and the accurate summary.

The organization of the manuscript is a bit weird, as it is written with the first part without a subheading, and then some subheadings. All the methods are put into the supplementary. I think there should be a distinct introduction, and there should be a shorter methods section, while it is OK to have a supplementary methods file with more details.

Thank you for pointing out the issues with the formatting. To address this comment, we have added the headings "Introduction" and "Results" in addition to subheadings within the Results section. While the Methods and Discussion sections are optional according to *Nature Communications* format requirements, we agree that including a brief Methods section would be helpful for readers. Therefore, we have added a concise Methods section in the main text, along with detailed Supplementary Methods.

I think the referral to plant microbiomics in both the first part and the discussion is a bit awkward. We do not need to explain to readers of *Nature Communications* that plants do not have pregnancies, birth and foster care. Apart from this the manuscript is well written and easy to follow.

Thank you for this comment. We have removed the comparisons to plants from the Introduction and Discussion sections. However, we decided to retain references to the experiments in plants that pioneered one-sided selection experiments.

- What are the noteworthy results?

The results are noteworthy, i.e. activity is to a wide extent microbiomically and not so much genetically driven. This is not fully novel, as there are several papers, some cited and some not cited in this manuscript, showing a strong microbiomic impact on behaviour, although the locomotor angle to a certain extent is new.

We completely agree that the finding that the gut microbiome can causally influence animal behavior is not, on its own, novel. The novelty of our study lies in experimentally demonstrating, for the first time in a mammalian system, that selection on a host trait can lead to changes in that same trait over time purely through microbiome transmission, without any genetic evolution in the host. This finding is supported by corresponding shifts in microbiome composition and metabolomic profiles, as well as by independent inoculation experiments. Together, these results provide experimental support for theoretical models (<https://doi.org/10.1146/annurev-biophys-101220-072829>) and highlight the potential for microbiome engineering in mouse models (<https://doi.org/10.1016/j.tim.2015.07.009>), advancing our understanding of ecosystem-level selection in complex mammalian microbiomes. We have now added this point to the discussion.

- Will the work be of significance to the field and related fields? How does it compare to the established literature? If the work is not original, please provide relevant references.

The work will be of significance to the field and related fields, as it very thoroughly documents the relation between microbiome and locomotor activity. Especially the transplantations are valuable, both because wild mouse microbiota is applied, and because such studies often do not do transplantations. Wild mouse microbiota is more diverse than the SPF microbiota, and, therefore, this is more relevant for humans, and it is probably also easier to reveal any differences. There could have been more references to microbiota works showing relation behaviour in mice, such as Bercik et al 2011. 'The Intestinal Microbiota Affect Central Levels of Brain-Derived Neurotropic Factor and Behavior in Mice', *Gastroenterology*, 141: 599-609.e3 and Bravo et al 2011. 'Ingestion of Lactobacillus strain regulates emotional behavior and central GABA receptor expression in a mouse via the vagus nerve', *Proc.Natl.Acad.Sci.U.S.A.*

Thank you for the references. We have added the suggested references to the discussion.

- Does the work support the conclusions and claims, or is additional evidence needed?

I think in general the conclusions are well supported.

Thank you.

- Are there any flaws in the data analysis, interpretation and conclusions? Do these prohibit publication or require revision?

I have difficulties understanding why Wilcoxon's Signed-Rank test is applied, as the data sets in it do not seem to be paired. In that case it should just be a Mann-Whitney U-test. Also, it is not clear why a non-parametric test is applied, because it is not described if the data were subjected to any normality and equality tests.

We appreciate the reviewer's comment regarding the use of the Wilcoxon test. To clarify, the data are not paired, and we used the **Wilcoxon rank-sum test** for pairwise comparisons. While this is sometimes referred to as the Mann-Whitney U-test, the two are mathematically equivalent.

We applied non-parametric tests because the majority of the 16 traits analyzed did not meet the assumptions of normality. Although a few traits were approximately normally distributed, we opted to use a uniform non-parametric approach across all traits for consistency and to take a conservative stance. This also allowed for more comparable interpretation of effect sizes and p-values across traits.

We revised the methods section to clarify our rationale, specify that data normality was assessed, and ensure consistent terminology.

- Is the methodology sound? Does the work meet the expected standards in your field?

It probably does, but we need more information on the animals (see below).

Please see the response above regarding the statistical test and the response below concerning the animals.

- Is there enough detail provided in the methods for the work to be reproduced?

It is not clear whether the intra-duodenal catheterization was done under anesthesia and if so, which one. It is also not clear whether analgesics were given afterwards. If not done under anesthesia and eventually some analgesics the procedure does not seem to be in accordance with the EU directive.

All experiments were conducted in accordance with NIH animal research guidelines and approved by the Institutional Animal Care and Use Committee (IACUC) of the Icahn

School of Medicine at Mount Sinai. We have added additional details in the Methods section in the main text and the supplementary methods.

Animals

Male C57BL/6J mice (Jax Strain #000664; Jackson Laboratory) were individually housed under a 12-hour light/dark cycle in a specific pathogen-free (SPF) facility. All mice were experimentally naïve, with no prior exposure to pharmacological agents or specialized diets, and exhibited normal health status at the time of experimentation.

Duodenal Catheterization Procedure

Mice aged 8–12 weeks (25–28 g) underwent duodenal catheter implantation. Preoperative analgesia (buprenorphine, 0.05 mg/kg, s.c.) was administered 30 minutes prior to induction of anesthesia with 3% isoflurane, followed by maintenance with 1.5% isoflurane. Animals were placed on a thermostatically controlled heating pad (CMA 450; Harvard Apparatus). The abdomen was shaved, disinfected with iodine soap, and sterilized with 70% isopropyl alcohol. A midline laparotomy was performed to expose the duodenum. A purse-string suture was placed 2 mm distal to the pylorus, into which a 3-mm segment of MicroRenathane tubing (0.025" OD × 0.012" ID; Braintree Scientific) was inserted. The tubing was anchored to the gastric antrum with an additional suture. The tubing was tunneled subcutaneously to the dorsal region and exteriorized through a small incision between the scapulae. The external catheter end was sealed until infusion procedures commenced. The abdominal wall and skin were closed using continuous and interrupted sutures, followed by topical application of Baytril ointment. Postoperative care included infrared heat-assisted recovery and intensive monitoring of locomotor activity and feeding behavior. Analgesia (buprenorphine, 0.05 mg/kg, s.c.) was administered twice daily for three days post-surgery.

The information on housing of the animals is very sparse, and we know nothing about enrichment, cage types, bedding etc.

Thanks for this comment. We have added information on enrichment, cage types, and bedding.

The lines were maintained in Individually Ventilated Cages (IVCs, Tecniplast) with a nestlet (Zoonlab), plastic hut or tubes (Datesand), and paper-based bedding (ALPHA-dri, SHEPHERD)...

What does it mean that health status was normal and how was this assessed? Normally, laboratory mice are free of a range of listed pathogens, but these mice were either of wild mouse origin and not subjected to rederivation, or they were germ-free, so even though I agree that the wild mouse derived mice are more natural, they do not seem very normal for laboratory animal facilities.

Although we did not explicitly state or directly measure the health status of the wild-derived house mice, we believe their health can be considered “normal” for several reasons.

First, the wild-derived mice used in the experiment had been maintained in captivity for over 20 generations through sib-sib mating from wild-caught founders and were never rederived (although rederived animals are available through Jackson Laboratory, <https://doi.org/10.1371/journal.pgen.1011228>). This inbreeding approach mirrors the standard process used to generate classical inbred mouse lines, and the fact that these animals were able to reproduce across multiple generations without visible health issues supports the argument that they are normal or healthy.

Second, key phenotypes (e.g. body weight, body length, and behavior) have been shown to reflect population-level differences across natural populations (<https://doi.org/10.1371/journal.pgen.1007672>, <https://doi.org/10.1111/mec.15476>), early-generation mice (N1 and N2s) (<https://doi.org/10.1371/journal.pgen.1007672>, <https://doi.org/10.1111/mec.15476>), and later generations (<https://doi.org/10.1371/journal.pgen.1011228>), representing natural variations in physiological traits.

Finally, natural variation in the gut microbiome is maintained in captivity for over 10 generations (<https://doi.org/10.1126/science.aat7164>), and several microbial lineages show evidence of vertical transmission in these mouse lines (<https://doi.org/10.1038/s41467-025-57435-z>), further suggesting a stable and representative biological system.

This has now been added to the supplemental text under “Wild-derived inbred lines reflect natural variation in host phenotypes and microbiomes”.

Germ-free is written as both germfree and germ-free. Correct MESH term is germ-free.

Thank you. We now use “germ-free” throughout the manuscript.

ILA is not properly defined in the text.

Thank you. ILA is now defined in the sub header and the text.

Reviewer #2 (Remarks to the Author):

In this original research, the authors test the hypothesis that phenotypic evolution of a host can be mediated by through microbiome transfer. The researchers first established a set of traits that were transferable via the microbiome and then chose one of these traits for a one-sided selection study to determine if host behavior could be modified through the microbiome. The research is an elegant, integrative study that is impressively replicated. The methods are sound, and in many cases exceed typical standards for work in this area. The results reveal at least one possible target (single microbe) of the selection experiments and delves deeply into mechanism through metabolite analysis and subsequent experiments. The research demonstrates that selection on the microbiome can significantly alter host phenotype independent of host genetics.

We thank the reviewer for their appreciation for this work.

Major questions/comments

My central questions/comments revolve around the phenotype of distance traveled. In the initial microbiome transfer experiments to the germ-free mice (results in Figure 1), how was it ascertained that it was the microbiome composition that altered the phenotype of the germ-free mice, and not the genetic background of the germ-free mice?

The germ-free mice used in our experiment were fully inbred C57BL/6 mice from Taconic, bred in our facility. As inbred mice are expected to have nearly identical genomes, we can confidently attribute the observed host trait changes to differences in microbiome treatment, rather than to genetic variation among the recipient germ-free mice. Any minor de novo mutations that may have occurred in our facility are accounted for by randomizing littermates into the two treatment groups.

If the reviewer is referring to potential interactions between the microbiome and alleles specific to the C57BL/6 background, our design still supports the conclusion that microbiome variation altered the host trait. However, this conclusion may be limited to the C57BL/6 genetic background, and further studies are needed to test microbiome–host genotype interactions. See the main text “While different host genotype–

microbiome combinations can yield distinct interactions ^{14,45,46}, the consistent phenotype transfer highlights the robustness of the microbial effect.”

Fig 1C suggests that germ-free mice, with either microbiome (MAN or SAR) travel more than the MAN line. How active are germ-free mice when germ-free? Are the results reported proportional to the baseline state of germ-free mice? Here, I am trying to understand how the microbial transfers changed the activity of the germ-free state?

We did not measure the distance traveled of germ-free mice maintained in sterile conditions in this study, so we cannot directly answer this question (but see comparison below). The reported values reflect germ-free mice after inoculation with microbiomes from two wild-derived mouse donors.

While we believe the main hypothesis can be tested without measuring distance traveled in germ-free mice under sterile conditions, we agree that such data would help in better interpreting our results.

Based on published studies, the distance traveled by our inoculated mice (200–400 m per 24 hours) is lower or comparable to reported values for germ-free mice in sterile conditions, which aligns with the known effect of the microbiome in reducing activity levels (<https://doi.org/10.1073/pnas.1010529108>). For example, using the same system (Promethion, Sable Systems), 12-week-old germ-free C57BL/6 mice in sterile conditions have been reported to travel ~460 m (<https://doi.org/10.3389/fendo.2019.00460>) to ~2500 m (<https://shorturl.at/gkiQ6>) in 24 hours. Our lower values are consistent with this trend and may also reflect the younger age at measurement (8-week-old), as locomotor activity tends to increase and peak around 12 months of age using the same measurement system (<https://doi.org/10.7554/eLife.72664>). We have added some of this point in the discussion.

“Although we did not measure distance traveled in germ-free mice under sterile conditions, the observed values in our study (200–400 m per 24 hours) are comparable to the published measurements for germ-free C57BL/6 mice using the same automated cage system (Promethion, Sable Systems) ⁷⁹.”

How much less active were the mice within the experimental treatments, and what was the differential between the selection lines and the control lines? That is, what was the effect size on overall movement? It looks like 10% by the end of the experiment. It would be useful for the reader to know the effect.

What is the fidelity of distance traveled? How much variation does an individual display among measurements?

Thanks for this comment. We have added the effect size (change in median distance traveled in meters: N0 vs N3 is -18.9% and N0 vs N4 is -9.8%) in the Discussion section. The overall fidelity is $\rho = 0.193$, $p = 0.0021$, as reported in the main text. Fidelity by each transfer round is shown in Fig. 2D&E and Fig. S4.

We only have one measurement per individual, so we are unable to assess within-individual variation. This is partly because behavior is a plastic trait, where multiple measurements could introduce variation unrelated to the microbiome, such as habituation to being moved from the original IVC rack to the temperature cabinet where we measure the behavior or the switch from IVC lids to Promethion lids.

Minor comments:

Pg 145 should this say “selection lines?”

Changed to “selection lines” and “control lines”.

What is plotted in Fig 2B, D? is it the mean of the 4 selection lines? Or individual performance? Please specify.

Fig. 2B is median as written in the figure legend, but we also added “for four selection lines and four control lines”. For Fig. 2D, we added “Data points represent individual donor-recipient pairs used in fecal microbiome transfer”.

Line 146 “or when using the raw data” what does this mean?

Clarified, “or when using the raw data uncorrected for these variables.”

Fig 3 F -what is the insert? It is too blurry to see what is in the insert.

It shows the positive relationship between NB value and distance traveled. We deleted the small panel within Fig. 3F (and Fig. 4D) and moved them to Supplemental Figures S7 and S11.

227- specify direction seems like this is “reduction” in distance traveled
Revised.

230- what does “this” refer to? The pattern? The outcome? Same for 233
Revised.

242: “top 3 negatively contributing taxa”. This is confusing, please revise
Revised.

344: Does Lactobacilli need to be italicized or written as lower case? What is the convention?

Lactobacilli is a common English plural and not a formal genus name. Thus, we revised it as *Lactobacillus* spp. following ASM nomenclature (<https://journals.asm.org/writing-your-paper#nomenclature>)

363: “microbes that persist across generations”- host generations?

Revised and clarified the argument: “Microbes that persist across host generations are particularly important, as they are more likely to drive coevolution compared to transient microbes, which may influence host traits over shorter ecological timescales.”

Reviewer #3 (Remarks to the Author):

The manuscript by Suzuki et al. provides valuable experimental evidence that microbiome selection alone can influence host traits in mammals, representing an important proof-of-principle for microbiome-mediated inheritance mechanisms. However, the central behavioral phenotype appears to rely heavily on statistical adjustments for confounding variables like body weight, with small effect sizes that required a high n and are visible after complex modeling rather than in raw data. While the microbiome and metabolite findings are compelling and the mechanistic follow-up with *Lactobacillus johnsonii* strengthens the work, the modest effect sizes and dependence on statistical correction of the behavioral data give me pause. With major revisions addressing the statistical robustness of the behavioral claims and clearer presentation of effect sizes alongside significance testing, this could represent an important contribution to our understanding of microbiome-mediated evolution.

Thank you for this thoughtful comment. We agree that the effect size is modest (change in median distance traveled: N0 to N3 = -18.9%, N0 to N4 = -9.8%), but we believe this does not diminish the significance of our study as the first proof-of-concept for microbiome-mediated inheritance of adaptive phenotypic plasticity in a mammalian model. Importantly, we present both raw data (Fig. 2B, upper) and two independent corrections (Fig. 2B, lower; Fig. 2C), which converge on the same result.

We corrected only two well-justified covariates: body weight at inoculation and batch effects. Body weight at inoculation (3–4 weeks) positively correlates with distance traveled (Fig. S2), and varies due to litter availability (Fig. S3). Batch effects, linked to litter size, season, animal care staff, time of day, and cage position, are known to

influence behavior. Given the inherent variability of behavioral traits, especially across batches, correcting for these factors is essential for reliable interpretation.

The modest effect size also reflects our conservative design. Unlike studies contrasting high and low lines, we compared selection and control lines, with the control line based on random selection, likely including low-activity individuals by chance (as you pointed out below). This approach, as recommended by critics of earlier one-sided selection studies (<https://doi.org/10.1016/j.tim.2015.07.009>), reduces power but improves rigor. Despite this, we observed significant differences with relatively small sample sizes and fewer selection rounds compared to previous studies in plants. These points are now clarified in the revised main and supplemental texts:

Main text: “Despite these improvements, the effect size of host trait changes in response to selection (change in median distance traveled in meters: N0 to N3 = -18.9%, N0 to N4 = -9.8%) was smaller compared to previous studies in plants^{13,25–27}. This may be partly due to our conservative design, in which the control line was established by random selection and happened to include low-activity individuals, thereby reducing the contrast in selection pressure between the selection and control lines”

Supplemental text: “One known factor that significantly influenced raw distance traveled (Fig. 2B, upper panel) was body weight (Fig. S2). Body weight at both the time of microbiome inoculation (3–4 weeks old) and the time of behavioral measurement (5–6 weeks old) positively correlates with distance traveled (Fig. S2). Because weight gain during this period is influenced by the microbiome, this supports the need to control for body weight only at inoculation (variation due to litter availability) but not at behavioral measurement (variation partly caused by microbiome differences) (Fig. S3).”

- When were microbiome samples analyzed?

The experiments were conducted in 2019 and 2020 (stated in supplemental methods), with microbiome analyses performed and re-analyzed between 2021 and 2025. The majority of final analyses reported here were conducted between 2024 and 2025.

- Why did you choose to use the open field test as a behavior marker of phenotype transmission?

To clarify, we did not choose the open field test for the one-sided selection experiment. In the initial experiment (Fig. 1), we assessed 16 traits, and selected the distance traveled, measured using the automated cage system Promethion (Sable Systems), as

the target trait for the one-sided selection experiment (Fig. 2). The open field test was used later as a gold standard behavioral assay to demonstrate that the microbial species and metabolites identified in the selection experiment independently influence mouse locomotor speed and distance (Fig. 5).

- It appears that a large difference in behavior is required in the donors (effect size 269%) to then pass to recipients (effect size of 15%). Does this indicate that only extreme differences may be transferable by microbiota transplant?

Thank you for this idea. We agree there may be a tendency for traits with large donor effect sizes to be more likely phenocopied in recipients, but we do not believe we have sufficient power to make that conclusion from the current data. Among the 16 traits assessed, only three showed significant transfer in the expected direction ($p < 0.05$), and six if we use a relaxed threshold of $p < 0.1$ (Fig. 1B). It is true that donor traits with large effect sizes, such as water intake (84.6%) and energy expenditure (25.3%), tended to be phenocopied in recipients (15.5% and 4.4%, respectively; $p < 0.1$). However, other donor traits with similarly large effect sizes, like fat mass (53.7%) and food intake (41.8%), were not transferred (recipient effect sizes 5.1% and 4.2%; $p > 0.1$). Additionally, some of these traits are highly correlated. A more definitive test of this hypothesis would require a study specifically designed to assess the relationship between donor effect size and microbiota-mediated trait transfer rates across a broader set of independent traits.

- Not quite sure I understand why you chose the SAR line as microbiome donors to reduce distance traveled, as they were the group with the higher distance traveled? Could you not have used the MAN donors to SAR recipients?

We used the SAR line as microbiome donors and germ-free C57BL/6 mice as recipients for the reasons outlined in the main text, including the fact that the SAR line exhibited higher distance traveled, which parallels natural scenarios where high-latitude mice with high activity disperse to low-latitude regions and evolve reduced activity. The use of germ-free recipients was also essential for conducting a one-sided selection experiment to isolate microbiome effects.

The reviewer's suggestion to transfer the MAN microbiome to SAR recipients to reduce distance traveled is well taken, and we have considered and attempted a similar approach. While antibiotic treatment of SAR mice is one option, it does not fully eliminate the endogenous microbiota. The cleanest approach would be to use germ-free SAR mice, which we attempted to establish with Taconic, but breeding issues made this unfeasible at the time.

- Why did you choose 2 mice at random for the control donors? If the random mice were either high or low in locomotor activity this could be a potential confounder.

Thank you for pointing this out. We selected two control donors at random, which likely included individuals with lower activity levels by chance. As we responded above, this approach was not only conservative but also intentional, based on critiques in the field that previous one-sided selection studies may have introduced bias without including random control donors (<https://doi.org/10.1016/j.tim.2015.07.009>). While this randomization could introduce noise and reduce our power to detect an effect, it allowed us to address concerns raised in the literature and strengthen the rigor of our design. The fact that we still observed a consistent and statistically significant response to microbiome selection under these conservative conditions is, in our view, a striking result, suggesting that the true biological effect may be stronger than the observed estimates.

- How was coprophagy microbiota transfers confirmed? It is difficult to assess dose and frequency using this method. It appears that some microbes were indeed transferred but it is difficult to say if this led to recipients having closely similar microbiome compositions to the donors.

This was confirmed in the first experiment, which showed partial but consistent transfer of distinct microbiomes through coprophagy (Fig. 1F). However, the reviewer is correct that oral gavage with a defined dose and frequency would likely have provided better control over the amount and composition of the microbiome transferred from donors to recipients. We chose coprophagy based on the observed fidelity (Fig. 1F) and to better mimic natural transmission, as one of the motivations of this study was to realistically test whether microbiome-mediated adaptive plasticity can be transmitted across generations.

- Single housing in itself is a stressor that can affect mouse behavior.

This is correct, and we did consider co-housing. However, we chose single housing for the following reasons. First, co-housing in microbiome experiments is often criticized due to pseudo-replication (<https://doi.org/10.1128/aem.01033-24>). This is because animals tend to share their microbiomes within the same cage, substantially reducing the number of independent biological replicates. Second, to minimize stress from single housing, we used only males, which are generally less sensitive to isolation stress than females (<https://doi.org/10.3390/brainsci10110799>, <https://doi.org/10.7554/eLife.18726>), and limited the duration to two weeks. Third, our measured serum corticosterone level (median 0.105 μ M, all data) falls within the baseline range reported for C57BL/6 mice ($0.08 \pm 0.01 \mu$ M) (<https://shorturl.at/GROF2>). Finally, and importantly, even if single housing introduced behavioral changes due to stress, all animals, control and selection

lines, were treated identically. Therefore, any potential stressor does not undermine the main conclusion of the study.

- It is not clear to me whether the behavior data in figure 2B is more driven by microbiota transfer or other variables. Did you expect the total distance traveled for control and selection to move in a similar direction? It looks like body weight was more different than the locomotor behavior, and it is well known that heavier mice typically move less. This makes me think that perhaps the microbes influence behavior through weight change rather than modulating other pathways that impact behavior.

Thanks for the comment. Yes, we did expect that control and selection lines would shift in a similar direction to some extent, as nearly all one-sided selection experiments to date show parallel fluctuations that cannot be solely attributed to selection (see ref 25-28 in main text). Although we took extensive measures to control for confounding variables, body weight was a known factor influencing distance traveled (Fig. 2B, upper; Fig. S2, Fig. S3). Because weight gain during the experiment is partly microbiome-driven, we corrected for body weight only at inoculation driven by litter availability. The reduced correlation in the body weight-corrected data (Fig. 2B, lower) suggests this covariate was successfully accounted for. This has been clarified in the main text and supplemental methods.

The reviewer's hypothesis that microbiome-driven behavior changes may be mediated through body weight is reasonable, but three lines of evidence argue against it. First, body weight at the time of measurement (5–6 weeks old) did not change significantly from the start to the end of the experiment for either control or selection lines (N0 vs N4, $q > 0.05$), despite a significant change in body weight at inoculation (3–4 weeks old) for the control line (N0 vs N4, $q < 0.05$), but not for the selection line (N0 vs N4, $q > 0.05$) (Fig. S3). This further justifies correcting only for body weight at inoculation. Second, no significant differences were observed between control and selection lines within any individual round (Student t-test, $p > 0.05$) (Fig. S3), suggesting a significant batch effect, which we also accounted for. Lastly, inoculation of *Lactobacillus johnsonii* and ILA into C57BL/6 mice altered behavior in the expected direction without affecting food intake or body weight (Fig. 5D, E, I, and J). These results support a role for the gut microbiome in modulating locomotor behavior independent of body weight changes.

We have incorporated this response into the Supplemented text by creating a new section called "Assessment of Body Weight as a Mediator of Microbiome-Induced Behavioral Change" along with a new supplemental Figure S3.

- Can you identify what the batch effects are?

In Table S27, we have added “batch” IDs to each animal, representing a combination of replicate number (e.g., R1, R2, R3) and transfer round (e.g., N0, N1, N2). Each batch corresponds to the set of multiple litters available at the time when the germ-free mouse cohort (which was always assigned to both control and selection lines) received fecal inoculation at 3–4 weeks of age. Biologically, we interpret batch effects as reflecting a combination of maternal environment (e.g., dam age, litter size, reproductive history), time of year (e.g., seasonal variation, changes in animal care staff), time of day (± 30 minutes), and cage position (even though positions were randomized). This explanation has been added to the supplemental text.

- It is difficult for me to see the difference in Bray-Curtis dissimilarity across control and selection groups. Can you explain what is being compared by the pairwise PERMANOVA p-value that is reported? I am concerned that the differences are being driven by a large n rather than biological relevance and that treatment explains much less variance than rounds of transfer.

Thank you. This concern highlights the main point we aim to convey: Treatment (control vs. selection lines) does not have a large effect on overall community structure, but the difference becomes statistically significant only at later rounds of transfer (we now emphasize this more clearly in the main text). In Fig. 3B, the PERMANOVA tests the overall effect of Treatment across all samples. In Fig. 3C, the pairwise PERMANOVA compares Bray-Curtis distances “within groups” (control-control and selection-selection) versus “between groups” (control-selection) within each transfer round. The confusion may have stemmed from how we plotted “within control” and “within selection” separately in Fig. 3C. We have revised the legend to clarify the comparison being made.

- Is locomotion speed necessary to include in figure 5? This is just distance divided by the standard 10 min test.

We believe presenting locomotion speed adds value for two key reasons.

First, EthoVision’s distance and speed measurements are not redundant, they provide distinct and complementary information. The software calculates velocity by dividing the distance traveled between frames by the time interval (<https://doi.org/10.3758/bf03195394>), repeating this process for each frame to generate a velocity profile across the test duration. If the two measures were redundant, one would expect the data points in Fig. 5B and 5C, as well as in Fig. 5G and 5H, to be identical apart from the Y-axis scale, which is clearly not the case.

Second, reporting speed in the open-field test enables more meaningful comparisons with the one-sided selection experiment (Promethion, Sable Systems), where only

movements exceeding a velocity threshold (1 cm/sec) are counted as distance traveled. As shown in Fig. 5, all animals moved at speeds above this threshold, supporting the validity of comparing the two systems.

We have added this rationale, along with details on how speed was calculated in the open-field test, to the supplemental methods.

- The study focuses on *Lactobacillus* but microbiome changes were complex - other taxa may contribute significantly. That being said, I appreciate that authors followed up initial findings and tested both one of the differentially abundant microbes and metabolites.

Thank you, and we completely agree. *Lactobacillus* and its metabolites emerged as top candidates from the one-sided selection experiment, and observing similar effects in a completely different microbiome background was surprising. That said, we also believe the underlying mechanism driving changes in distance traveled is likely complex, as discussed in the main text:

“Although this provides a mechanistic hypothesis for the targets of selection within complex microbial communities, the underlying mechanism is likely more complex, involving many additional microbial taxa and metabolites contributing to the observed outcome ⁵.”

- This is essentially a single experiment (though with internal replication). Independent replication would strengthen conclusions.

Thank you for the comment. We are testing the same main hypothesis with a single starting microbiome. However, after NO, each control and selection line never exchanged microbiomes and can be treated as independent replication (<https://doi.org/10.1128/aem.01033-24>). We believe the experimental design improves upon previous studies and that the conclusions are robust. That said, we agree that repeating the experiment with multiple starting microbiomes and continuing the experiment into further rounds of transfer would strengthen the conclusion as mentioned in the discussion.

- Why were the metabolomic analyses focused to 12 targeted compounds?

Targeted and non-targeted metabolomics each have trade-offs. Non-targeted metabolomics allows the discovery of unexpected or novel metabolites associated with distance traveled but suffers from challenges such as multiple hypothesis testing and lower reliability in metabolite identity and concentration due to the lack of calibration. In contrast, targeted metabolomics, focusing on 12 metabolites previously reported to be

involved in the gut-brain axis (<https://doi.org/10.1128/mbio.02836-23>), is more hypothesis-driven, provides greater confidence in the identity and quantification of metabolites, and facilitates mechanistic insights grounded in prior literature. While we acknowledge that a targeted approach may miss other relevant metabolites, the primary aim of this study is to test whether microbiome selection and transmission can pass on microbiome-mediated traits, independent of metabolic findings.

- Were all data subjected to FDR testing?

Yes, we reported both uncorrected and FDR-corrected p-values for transparency. When uncorrected p-values are shown, we explicitly state it. Unless otherwise noted, all reported p-values are FDR-corrected.

- Were there any outliers removed?

Yes, as described in the Methods section: "The samples outlying (outside of median \pm 3 sd) by body weight at inoculation (BWi) or distance traveled were discarded, leaving 303 samples. For the analyses involving metabolomic data, three samples outlying by their metabolomes were also excluded from consideration (leaving 118 samples with both metagenomes and metabolomes)."

- In Figure 1B both dark yellow and light grey are referred to as $P < 0.01$. Please define what the gray boxes represent in the figure or figure legend.

Thank you. The dark yellow ($P < 0.01$) and light grey ($P < 0.1$) indicate different significance cut-offs, but we agree it is confusing since the grey includes all P-values below 0.1. We have revised the legend for clarity (e.g., $P = 0.05-0.1$).

- The use of "non-nuclear" in the one sentence summary is somewhat confusing terminology. I think I understand what the authors are asserting but perhaps something along the lines of "extra-genomic mechanism" or "microbiome-mediated inheritance" would be easier for readers to understand.

Thanks for the suggestion. Here is the revised one sentence summary: "We experimentally demonstrate that host traits under selection can be transmitted solely through the microbiome, highlighting microbiome-mediated inheritance as a mechanism shaping host ecology and evolution."

REVIEWERS' COMMENTS

Thank you for all the valuable comments and for re-reviewing our manuscript. Our responses are in blue.

Reviewer #1 (Remarks to the Author):

In general, the authors have followed my advice, and we have also had more information on the animals. I have no further comments.

Thank you. The key details on animal housing and the point about the health status of wild-derived inbred mice strengthen the manuscript.

Reviewer #2 (Remarks to the Author):

Thank you for your responses to my comments. I realized my first comment was not as clear as it could have been. It is reproduced below::

In the initial microbiome transfer experiments to the germ-free mice (results in Figure 1), how was it ascertained that it was the microbiome composition that altered the phenotype of the germ-free mice, and not the genetic background of the germ-free mice?

What I was attempting to ask was how is it known that it was the different microbiomes that caused the differences in phenotype and not just the genetics of the host (or a potential interaction with host genes). Since the comparison was between hosts with different microbiomes, and not a comparison to germ-free mice, how are the authors certain that the phenotype is caused by the microbiome per se and not the underlying genetics of the host. That does the phenotype differ from the germ free state?

Thank you for clarifying your comment. Whether the phenotypes of germ-free mice inoculated with a microbiome differ from those of germ-free mice maintained in sterile conditions is a valid question. However, the purpose of our initial experiment was to identify host phenotypes that are most influenced by differences between two microbiome communities. Our goal was to select a host trait mediated by microbiome differences, rather than by the contrast between germ-free and non-germ-free conditions, to motivate the main selection experiment. For example, even if germ-free mice colonized with the two microbiomes did not differ significantly from germ-free mice in sterile conditions (e.g., values between SAR vs. MAN recipients), we still observed a significant difference in phenotype between SAR and MAN recipients themselves (Fig. 1E). Thus, regardless of comparisons with germ-free mice in sterile conditions, we would have chosen distance traveled as the focal trait for the main selection experiment.

That said, the reviewer is correct that we did not directly demonstrate differences between phenotypes of germ-free mice maintained sterile versus those inoculated with microbiomes under our specific facility and housing conditions. However, there is strong indirect evidence that the observed phenotypes are likely caused by microbiome inoculation. (1) As noted in our original response, the distance traveled by our inoculated mice (median 250–300 m per 24 hours, Fig. 1E) is comparable or lower than reported values for germ-free mice in sterile conditions: using the same system (Promethion, Sable Systems), 12-week-old germ-free C57BL/6 mice in sterile conditions have been reported to travel ~460 m (<https://doi.org/10.3389/fendo.2019.00460>) to ~2500 m (<https://shorturl.at/gkiQ6>) in 24 hours. (2) Reduced activity levels of inoculated germ-free mice compared to sterile germ-free mice have been documented since 2011 (<https://www.pnas.org/doi/full/10.1073/pnas.1010529108>).

We believe the revised response in the discussion will further address the reviewer's comment:

“...although we did not measure distance traveled in germ-free mice under sterile conditions, the values observed in the initial (median 250-300 m per 24 hours, Fig. 1E) and selection experiment (median 180–250 m per 24 hours, Fig. 2B) are similar to or lower than the published measurements for germ-free C57BL/6 mice using the same automated cage system (Promethion, Sable Systems)⁷⁹. Reduced activity levels of inoculated germ-free mice compared to sterile germ-free mice are consistent with previous reports⁸⁰.

Reviewer #3 (Remarks to the Author):

Despite my initial concerns about modest effect sizes and statistical dependence, the authors adequately justified their conservative experimental design and demonstrated that effects are visible in raw data, not just after corrections. The independent validation through *Lactobacillus* experiments and the rigorous use of random controls (rather than extreme phenotypes) strengthen the biological interpretation. While independent replication would be ideal, this represents the first experimental demonstration of microbiome-mediated trait inheritance in mammals, a significant conceptual advance regardless of effect magnitude. This work opens exciting new avenues for understanding how microbial communities can drive evolutionary change and may have important implications for microbiome engineering approaches. I recommend to accept this manuscript for publication.

Thank you. Addressing your comments on effect size has significantly improved the strength, conservative design, and novelty of this study.